



# Multi-centuries mean summer temperature variations in the Southern Rhaetian Alps reconstructed from *Larix decidua* blue-intensity data

Riccardo Cerrato[1], Maria Cristina Salvatore[1, 2], Michele Brunetti[3], Andrea Somma[1], Carlo Baroni[1, 2]

[1]Earth Sciences Department, University of Pisa, Pisa, 56124, Italy
[2]Geosciences and Earth Resources, National Research Council of Italy, Pisa, 56124, Italy
[3]Institute of Atmospheric Sciences and Climate, National Research Council of Italy, Bologna, 40129, Italy

*Correspondence to*: Riccardo Cerrato (riccardo.cerrato@unipi.it)

**Abstract**

Ongoing climate change is likely to cause a worldwide temperature increase of 1.5 °C by the mid-century. To contextualize
these changes in a long-term context, historical climatological data extending beyond data obtained from instrumental records are needed. This is even more relevant in remote areas characterized by complex climatic influences and where climate sensitivity is pronounced, such as the European Alps. Dendroclimatology has been recognized as a fundamental tool for reconstructing past climate variations because its temporal resolution is higher than that of other proxies. In this study, we present a comprehensive dendroclimatic analysis in which blue intensity (BI) data derived from European larch (*Larix decidua*
Mill.) trees in the Southern Rhaetian Alps were employed. By establishing the relationships between BI patterns in tree rings and climate variables, the possibility of using the obtained data for constructing a high-resolution, long-term climate record is explored. The results showed that BI data from European larch share greater variance with June–August mean temperatures than total ring width measurements. Moreover, the BI performance as a temperature predictor resulted temporal and spatial quite-stationarity and its regression indices are comparable to those obtainable by data from the more expensive wood density
method. The results from this analysis will extend the current knowledge on the applicability of using BI data to study European larch, and the reconstruction described herein is the first attempt to determine whether this proxy can be utilized for dendroclimatic aims. Thus BI data represent a new tool for extending our knowledge beyond that obtained from instrumental records and facilitating a more robust evaluation of climate models and future climate scenarios.

## 1 Introduction

Climate change has been recognized as unequivocally induced by human activities (IPCC, 2023; Eyring et al., 2023), and it is extremely likely that this activity has been the dominant cause of the observed warming since the mid-20[th] century (IPCC, 2013). Although human-induced global warming is likely to cause a worldwide temperature increase of 1.5 °C between 2030 and 2050 CE (IPCC, 2018), its effects on high-altitude areas are even more emphasized, with a temperature increase rate that is almost double that of the global mean (Pepin et al., 2015; Brunetti et al., 2009; Auer et al., 2007; Böhm et al., 2001). This
enhanced warming rate implies not only an accelerated glacier melting, reduced snow cover duration, and permafrost thawing



but, as a consequence, disruption of the hydrological cycles, disturbance of terrestrial and freshwater species and ecosystems, slope instability and a greater probability of wildfires (Carrer et al., 2023; IPCC, 2019a). Understanding the dynamics of climate variability over centuries has been not only a scientific endeavour but also a pressing concern for society at large, as it provides critical insights into the Earth's response to natural and anthropogenic factors (IPCC, 2023, 2019a, b, 2018). Thus, to

contextualize ongoing climate and global changes in a wider frame, precise information on past environmental and climatic conditions is needed.

Long-term and validated meteorological instrumental time series are the best tools for studying and analysing the climate of the past, but these data are not spatially homogeneous and are rarely available for remote sites. Moreover, their time span can also represent a limitation. For instance, the European Alps are among the areas in which long-term meteorological

instrumental time series exist, covering at least the last two centuries (Brunetti et al., 2012; Auer et al., 2007; Brunetti et al., 2006). However, the number of meteorological stations, and thus their representativeness of the high-elevation and remote regions in the inner alpine valleys, is much lower before 1875 CE (Brunetti et al., 2006). Thus, the use of proxies capable of representing meteorological variables before the 19[th] century is essential for better understanding the changes that involve high mountain areas.

Among the climatic proxies that can be used (Trachsel et al., 2012), dendrochronology represents an excellent tool for reconstructing the climatic variations that occurred in the past. In fact, tree-ring-based dendroclimatology has emerged as a powerful tool for reconstructing past climate variability, offering a unique perspective on long-term environmental changes at both hemispherical (Esper et al., 2018; Anchukaitis et al., 2017; Wilson et al., 2016) and regional (Büntgen et al., 2011; Corona et al., 2010; Büntgen et al., 2006) scales at an annual resolution. However, these large-scale reconstructions are dependent on

local data that are also useful for performing reconstructions at the local scale and thus highlight local climatic patterns as well. In this context, the Alps are an important site, representing a hinge among a continental climate that characterizes central Europe, a Mediterranean climate that characterizes southern Europe, and a more Atlantic climate that is present in the westernmost portions of the European continent. In this context, the Southern Rhaetian Alps, hosting the southernmost glaciers of the central Alps, present an intriguing region for dendroclimatic investigations, as demonstrated by previous studies

performed in the area (Unterholzner et al., 2024; Cerrato et al., 2023, 2020, 2019a, b, 2018; Coppola et al., 2013, 2012; Leonelli et al., 2011).

Traditionally, dendroclimatic reconstructions have relied on measuring the annual total ring width (TRW) of trees. However, the quest for more robust and high-resolution climate records that are less affected by growth trend problems has led to the exploration of innovative methods such as maximum wood density (MXD), anatomical traits, and isotopes (Leavitt and Roden,

2022; Björklund et al., 2019). Among these, blue intensity (BI) data have emerged as a promising tool, offering a potential alternative to overcome the costs of MXD analysis (McCarroll et al., 2002) and allowing a greater number of laboratories to perform MXD-like analysis (Reid and Wilson, 2020; Wilson et al., 2017b, 2014). In fact, the BI data, derived from the spectral analysis of tree-ring samples, provide climatic information that is virtually identical to that acquired through MXD in terms of nonlinearity, temperature correlation strength, and autocorrelation (Ljungqvist et al., 2020); these data are a function of the



cell wall dimension rather than the TRW or cell wall compound (Björklund et al., 2021). However, even if BI and MXD data are comparable, differences between the proxies could emerge as a function of the intrinsically different resolutions of the two methods (Björklund et al., 2019), as is also underlined by the anatomical MXD (Seftigen et al., 2022; Björklund et al., 2020). Blue intensity, albeit a relatively new methodology, has already been tested on several coniferous species (e.g., Scots Pine (*Pinus sylvestris* L.) in Fennoscandia and Scotland, various *Picea* ssp., *Pinus* ssp., and *Tsuga* ssp. in North America and Europe

and other coniferous species in Asia and Oceania (see Reid and Wilson, 2020; Cerrato et al., 2023 for more information). Nevertheless, in the European Alps, only a few studies have been performed using BI data (Cerrato et al., 2023; Arbellay et al., 2018; Nicolussi et al., 2015; Österreicher et al., 2014), and even fewer have been performed on European larch. Thus, additional tests are needed (Reid and Wilson, 2020). The European larch (*Larix decidua* Mill.), the dominant tree species in the Southern Rhaetian Alps, is particularly well suited for dendroclimatic investigations due to its sensitivity to environmental

conditions and longevity, holding great potential for dendroclimatic studies (Cerrato et al., 2018; Coppola et al., 2013, 2012; Büntgen et al., 2011, 2006). Although this species has been widely studied using both TRW and MXD, the associated BI data were used only for dendroentomological aims (Arbellay et al., 2018), and a dendroclimatic analysis of the BI data from European larch is still lacking.

This paper aims to present a comprehensive dendroclimatic analysis and climatic reconstructions utilizing European larch BI

data from samples collected in the Southern Rhaetian Alps. The methodology, advantages, and limitations of using BI data in the context of climate reconstructions are analysed, and the relationships between BI data and climate variables are examined with the aim of constructing a high-resolution, long-term record of climate variability.

## 2 Study area

The study area is located on the Adamello–Presanella and Ortles-Cevedale massifs (Southeastern Alps, Southern Rhaetian

Alps, Marazzi, 2005). The area is characterized by a high number of peaks exceeding 3000 m a.s.l. (e.g., Vióz Mount – 3645 m; Adamello Mount – 3554 m; and Presanella Mount – 3558 m) and is one of the most glaciated and glacierized areas in the Italian Alps (Salvatore et al., 2015). The sampling stands (Bosco Antico – ANBO – on Ortles–Cevedale; Val di Barco – BARC – and Pradach di Val Palù – PALP – on Adamello–Presanella Massif) belong to treeline ecotones and span between 1820 and 2270 m a.s.l. with a general northern exposure (Fig. 1). European larch individuals are scattered in an environment where

ericaceous species are predominant (*Rhododendron ferrugineum* L., *Vaccinium* spp.) (Andreis et al., 2005; Gentili et al., 2013, 2020). Soils are commonly influenced by parent material and superimposed vegetation and can be classified as immature and shallow podzols, histosols or umbrisols (IUSS Working Group, 2007; Galvan et al., 2008).



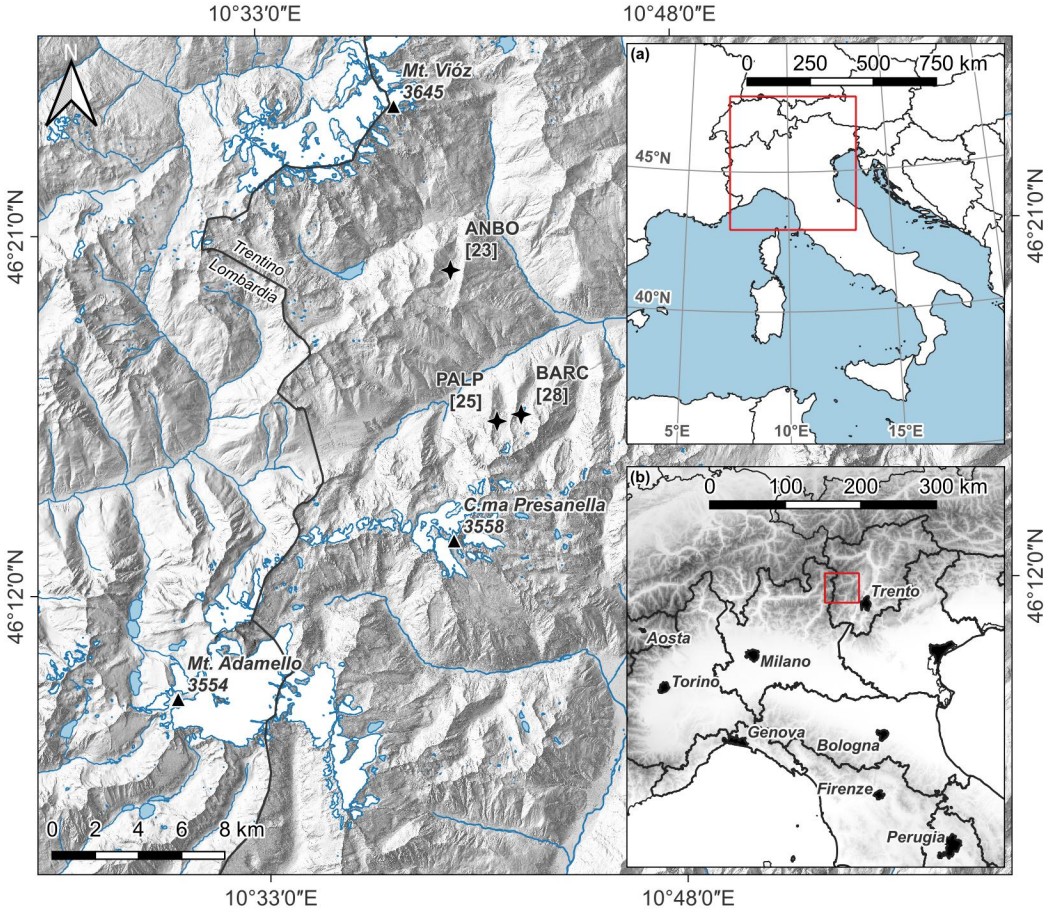

**Figure 1: Map of the study area and sampling sites. Stars indicate the sampling sites. Numbers within square bracket indicate sample size. Inset (b) base map: European Union Digital Elevation Model (EU-DEM). The red squares in insets (a) and (b) represent the footprint of inset (b) and the main map, respectively.**

The area is characterized by a latitudinal precipitation pattern that shows a decreasing trend northwards and is located just south of the so-called "inner dry alpine zone" (Isotta et al., 2014). The precipitation distribution reaches a minimum in winter and a maximum in summer, at 140.8 mm and 288.1 mm, respectively, whereas the mean annual value is 928.4 mm (Crespi et al., 2018; Carturan et al., 2012). Considering the temperature, the 1961–1990 mean annual temperature measured at the nearest station (Careser meteorological station ca. 12 km northwards from the sampling stands and located at 2607 m) was –1.2 °C, with February representing the coldest month (–8.3 °C) and July the warmest month (+6.9 °C).



## 3 Materials and methods

### 3.1 Tree-ring and blue intensity data

In this study, cores from 76 European larch trees were sampled over the past decade and prepared for total ring width (TRW) measurements (for sampling, sample preparation, measurement, and cross-dating details, refer to Cerrato et al. 2018, 2019b). The collected samples were repolished with progressively finer sandpapers with up to P2000 grit to remove pencil marks and to highlight the ring boundaries, then they were scanned at 3200 dpi using a flat-bed scanner Epson Perfection V850 Pro (Seiko Epson Corporation, Suwa, Japan) with SilverFast Archive Suite 8 software (LaserSoft Imaging AG, Kiel, Germany). The

scanner acquisition colours were calibrated using an IT8.7/2 colour card. BI measurements were subsequently performed using CooRecorder 9.5 Software (Cybis 2020 – http://www.cybis.se/forfun/dendro/index.htm).

The settings of the frame for calculating the BI value can vary depending on the species, site and scientific purpose (Buckley et al., 2018; Dannenberg and Wise, 2016; Schwab et al., 2018; Kaczka et al., 2018; Rydval et al., 2014; Tsvetanov et al., 2020). In this study, a frame width of 100 pixels was used to measure the minimum latewood BI (LWBI) and maximum earlywood

BI (EWBI) values, whereas frame depths of 50 and 200 were used for measuring the LWBI and EWBI, respectively. The offset of the frame was set at 5 and –2 for the LWBI and EWBI, respectively (Fig. S1 in the Supplementary Material). For both the LWBI and EWBI, we considered the mean values of the 25 %, 50 %, 75 % and 100 % of the darkest (and lightest when considering the EWBI) pixels in the frame (Cerrato et al., 2023). For easier comparison with climate data, BI values were inverted following standard procedures (Rydval et al., 2014; Wilson et al., 2014).

Extractives and wood discolouration are other issues encountered in BI studies that devise different solutions on the basis of species, site, and scientific purpose (Solomina et al., 2016; Fuentes et al., 2018; Sheppard and Wiedenhoeft, 2007; Wilson et al., 2017a). Following Arbellay et al. (2018), in this study, the extractives were not removed; however, to correct the heartwood/sapwood discolouration that characterizes European larch, 16 Delta BI (DBI) datasets were calculated and analysed as differences between the EWBI and LWBI datasets (Björklund et al., 2015, 2014).

The obtained BI sample series were visually and statistically cross-dated with the TRW series to check the correctness of the results. BI sample series belonging to the same individual were averaged to create the individual BI series. Some individual BI series showed an age trend; thus, they were standardized using a modified negative exponential curve. If the modified negative exponential curve failed to fit the trend of the individual series, they were standardized using a negative or a horizontal line. Following the first standardization, to highlight the high-frequency domain (*sensu* Melvin 2004) and attenuate the

discolouration bias in EWBI and LWBI, a high-pass Gaussian filter with a window length of 30 years and a sigma of 5 years was applied to the individual series. The mid-low-frequency domain was obtained using the same filter as a low-pass filter. Only those series that showed high correlation values with the site master chronology (i.e., p-value less than 0.001 and Spearman's ρ greater than or equal to 0.30) were considered to construct the site chronologies.

Site chronologies were obtained as a biweighted robust mean of the series belonging to each site where the variance was

stabilized as a function of the sample depth (Schweingruber, 1988; Fritts, 1976). The expressed population signal (EPS) was





calculated to estimate the representativeness of the sampling compared to an infinite hypothetical population, considering the uneven sample depth (Fritts, 1976), and the commonly used threshold of 0.85 was used to limit the site chronologies in time.

To highlight the common signals that characterize the three site chronologies, principal component analysis (PCA) and evolutionary principal component analysis (EPCA) were performed (Camiz and Spada, 2023; Camiz et al., 2020). This

approach limits the period of analysis to the shortest considered chronology but allows the retention of only those factors that explain the *a priori* decided quantity of the original data variance. In this study, the components that explained 80 % of the variance in the original dataset were considered.

All the data were manipulated in the R-project environment (R Core Team, 2022) using dplR (Bunn, 2008, 2010) and smoother (v. 1.1, https://CRAN.R-project.org/package=smoother, accessed on 09 October 2023) packages, whereas the principal

component analysis was performed using the 'stat' and FactoMineR (Lê et al., 2008) packages.

## 3.2 Instrumental data

Instrumental series for minimum, maximum and mean temperature and for precipitation were considered to explore the sensitivity of the BI chronologies to climate variability. Meteorological series from 1764 to 2015 specific to the sampling stands were reconstructed by interpolating the climate information provided by meteorological station data using the anomaly

method (New et al., 2000; Mitchell and Jones, 2005) and interpolating the longest and homogenized meteorological series available for the Alpine region (Brunetti et al., 2006, 2012, 2014; Crespi et al., 2018). The interpolation procedure consists of the independent reconstruction of the climatologies (i.e., the climate normals over a given reference period) and the deviations from them (i.e., anomalies).

Climatologies, linked to geographic features of the territory, are characterized by large spatial gradients; anomalies, linked to

climate variability and change, are generally characterized by greater spatial coherence than climate normals.

Therefore, the former were reconstructed by applying an interpolation technique that exploits the local dependence of meteorological variables on elevation (Brunetti et al., 2014; Crespi et al., 2018). This technique requires a high spatial density station network, even if the data are available for a limited period only.

On the other hand, anomalies can be reconstructed through a simpler interpolation technique and a lower station density.

However, long temporal coverage is mandatory, as is accurate homogenization of the time series for removing nonclimatic signals (e.g., due to instrument relocation and changes in measurement practices).

Finally, from the superimposition of climatologies and anomalies, monthly temperature and precipitation series in absolute values that were representative of the specific sites were obtained.

Information about the techniques and their accuracy are provided in Brunetti et al. 2012, 2014 and Crespi et al. 2018.

In addition to the site-specific series reconstructed as described above, the Climatic Research Unit (CRU) time series (TS) version 4.07 of a high-resolution spatially continuous interpolated gridded dataset (Harris et al., 2020) was accessed through the CEDA archive (https://data.ceda.ac.uk/badc/cru/data/cru_ts/cru_ts_4.07, last accessed 10 October 2023) to assess the





spatial coherence of the dendroclimatic signal. The meteorological data were filtered as BI data to minimize bias in correlation with the tree-ring data resulting from unfiltered trends (Cerrato et al., 2018, 2019a; Seftigen et al., 2020; Ols et al., 2023)

## 3.3 Climate correlation and reconstruction

The BI–climate relationship was assessed by calculating Pearson's correlation indices between the BI chronologies and climatic parameters. The correlation indices were calculated over the 1764–2013 period considering the monthly values from the previous May to October or the current year of growth, for a total of 18 months. In addition, considering the climatic sensitivity of the European larches in the area (Coppola et al., 2013; Cerrato et al., 2018), seasonal variables were created and examined by averaging the temperature of the June–August (JJA) period. The stability of the BI-climate relationship over time was verified through moving correlation analysis performed for each year using a window width of 50 years. Correlation indices were calculated in the R-project statistical environment by means of the treeclim (Zang and Biondi, 2015) package applying the bootstrapping procedure described in DENDROCLIM2002 (Biondi and Waikul, 2004).

To test the stationarity of the transfer function and thus to assess the reliability of the reconstruction (Wilmking et al., 2020), a bootstrapped cross-calibration-verification (CCV) approach was applied, and the reduction in error (RE) and the coefficient of efficiency (CE) were calculated (Fritts, 1976; Cook et al., 1994). Moreover, the bootstrapped transfer function stability test (BTFS) was performed (Buras et al., 2017). The process was repeated 1000 times. Dendroclimatic reconstruction was assessed by performing linear regression between z-scores of both BI values (predictor) and meteorological data (predictand) considering an ordinary least-square regression approach. Then, the mean and the variance of the DBI data were adjusted against the instrumental targets to avoid the typical loss of amplitude due to regression error (Carrer et al., 2023).

## 4 Results

From the 76 sampled trees, 24 BI series (i.e., four EWBI, four LWBI and 16 DBI series) of 24248 rings were obtained for a maximum period of 599 years (i.e., 1418–2016 CE; Table 1). Considering the EPS, the remaining chronologies spanned between 1502 and 2015 CE for a total of 514 years, whereas the mean interseries correlation increased by 0.05 at most (Pearson's $\bar{r}$, Table 1).

| valley | Time span [total years] EPS* span [total years] | n. of trees [used] | $\bar{r}$ [EPS* $\bar{r}$] $\bar{\rho}$ [EPS* $\bar{\rho}$] | correlation with other chronologies |
|---|---|---|---|---|
| ANBO | 1418–2015 [598] 1502–2015 [514] | 23 [19] | 0.51 [0.54] 0.50 [0.53] | BARC: 0.68 PALP: 0.70 |
| BARC | 1502–2016 [515] 1730–2013 [284] | 28 [18] | 0.43 [0.45] 0.40 [0.40] | PALP: 0.77 – |



| PALP | 1566–2015 [450] | 25 [22] | 0.49 [0.54] | – |
| | 1611–2015 [405] | | 0.51 [0.54] | – |
| *PCA (ANBO+BARC+PALP)* | *1730–2013 [284]* | *76 [59]* | | |
| *PCA (ANBO+PALP)* | *1611–2015 [405]* | *48 [41]* | | |

**Table 1: Statistical parameters of the resulting BI chronologies and their Pearson ($\bar{r}$) and Spearman ($\bar{\rho}$) mean interseries correlation coefficients. *EPS stripped chronology as described in Sect. 3.1.**

The analysis of the 24 BI chronologies and their correlation with meteorological datasets revealed that the influence of the

considered percentage of the LWBI darkest pixels, as well as that of the EWBI lightest pixels, slightly affected the results. However, in general, DBI values obtained from a small percentage of the darkest LWBI pixels (i.e., 25 %) and from a large percentage of the EWBI pixels (i.e., 100 %) yielded better results considering the values of their correlation with meteorological data (not shown). Thus, for further analysis, the series obtained from the difference between 25 % of the darkest latewood pixels BI and 100 % of the earlywood pixels BI were used and identified as DBI.

The DBI site chronologies showed a similar trend, with only the BARC reporting a slightly positive upwards trend over time (Fig. 2). Moreover, the maximum sample depth of each chronology was comparable in quantity but not in time, with the duration reduced by approximately 100 years between ANBO and PALP and another 100 years between PALP and BARC. Due to the coherence shown by DBI chronologies, a PCA was performed considering all the chronologies to better preserve the common variability, likely related to climate (Seftigen et al., 2020). Moreover, the use of PCA permits us to obtain a

slightly better correlation coefficient when correlated with meteorological time series (not shown).

The PCA confirmed that the considered chronologies shared a large portion of the original variance. In fact, the first principal component (PC1) explained more than 80 % of the variance alone, and all the chronologies were strongly positively correlated with it. Thus, PC1 (ANBO+BARC+PALP) was selected to represent the areal chronology (Fig. S2 in the Supplementary Material for further details).

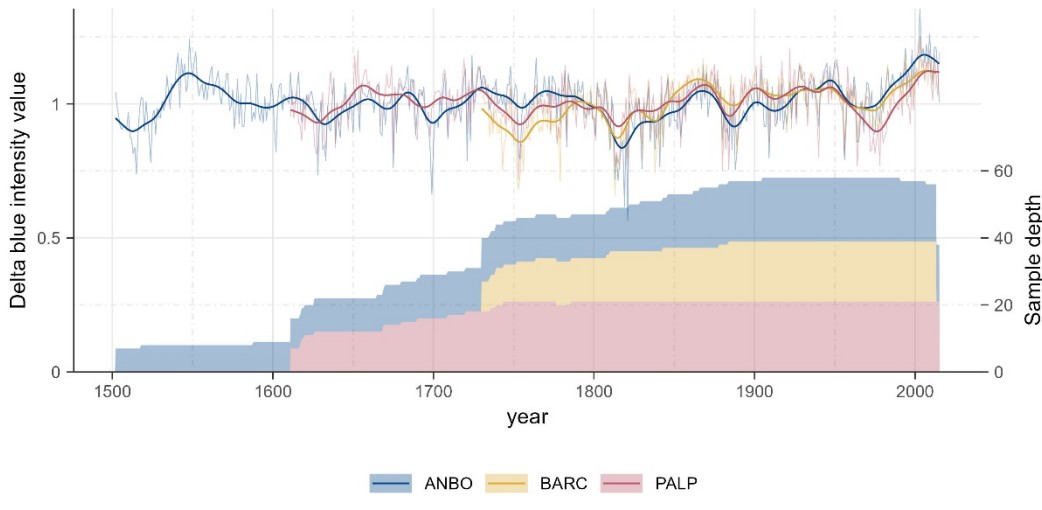




**Figure 2: Delta blue intensity (DBI) chronologies and sample depths. The thick solid lines represent 30-year Gaussian-filtered chronologies (σ = 5 years).**

## 4.1 BI–climatic correlations

PC1 (ANBO+BARC+PALP) was correlated with the meteorological datasets. Higher correlation values were obtained with
the mean temperatures (Fig. 3), whereas the mean minimum and maximum temperatures returned similar but lower values
(not shown). Precipitation was also tested, but the correlations were not significant.

Regarding the static correlation of the raw data (Fig. 3a), PC1 (ANBO+BARC+PALP) exhibited significant positive
correlation values with temperature of previous summer and fall seasons (i.e., May–November) and those of current late spring
and summer seasons (i.e., May–September), with the former returning lower values than did the latter. The highest correlation
value, however, was obtained with the mean temperature during JJA (0.68, 95 % confidence interval: 0.58–0.75). Regarding
the low-frequency domain, the correlation was always significant and positive, with the highest values occurring in summer
(June–August), both previous and current. Even in the low-frequency domain, the highest value was obtained for the current
JJA (0.78, 0.71–0.83). For the high-frequency domain, lower values than those of the raw and low-frequency domains were
obtained. The correlation was slightly significant for the previous October ($\alpha<0.05$) and for the current summer (June–August).
Like in previous domains, at high frequencies, the highest correlation results were also obtained for the mean temperature
during JJA (0.56, 0.45–0.66).

Regarding the moving correlation values obtained between the JJA mean temperature and both the raw data and high-frequency
domains (Fig. 3b), a high degree of stationarity can be observed. In fact, the correlations are characterized by high and stable
coefficients (always $\alpha<0.01$ but $\alpha<0.001$ for most of the considered windows). Moreover, the raw data did not show a
statistically significant trend, in contrast to the high-frequency domain, which is characterized by a negative trend in correlation
coefficients (Mann–Kendall S: –7705, p-value<0.001) associated with an increase in variance. Moreover, both correlation
series showed a decreasing step in the right-aligned 1964 CE window. In contrast to the raw data and high-frequency domain,
the low-frequency domain exhibited greater instability, especially in the 1880s–1930s and 1960s–1980s periods, but with
generally more limited variance.

The high correlation values obtained between PC1 (ANBO+BARC+PALP) and the JJA mean temperature were also
corroborated by good visual agreement between the time series and generally high statistics, with BI values that can explain
between 31 % and 61 % of the temperature variance as a function of the considered domain (Fig. 3c).

The spatial correlation between the raw BI and JJA mean temperature data returned positive values which significantly covered
all central and southern Europe, North Africa, and the Middle East (Fig. 4a). However, the spatial correlation is not stable over
time. In fact, after an initial decrease (until the 1965 CE window), high and significant correlation values were concentrated
around the Mediterranean basin until the beginning of the 2000s. Since then, a strong increase in the raw spatial correlation
was observed (Fig. S3 in the Supplementary Material).



Considering the spatial representativeness of the high-frequency domain, it was more limited than the raw data but still strongly and positively correlated with a large portion of the western and central Mediterranean basin (i.e., from 20° W to 25° E) to

central Europe (circa 53° N) (Fig. 4b). Regarding the stability over time, significant values decreased from an area well established across the central and western Mediterranean and Europe (1955 CE window) to an area that involved mainly Italy, the western Mediterranean, and Algeria in the 2010s (Fig. S4 in the Supplementary Material).











**Figure 3: Pearson's static correlation coefficient between DBI PC1 and mean temperature for the period 1775–2013. The coloured**
**bars indicate that the correlation values are significant at least at the 0.05 level. The solid black vertical line indicates the 95 %**
**confidence interval. All capitalized month abbreviations indicate the current year (a). Pearson's moving correlation coefficient (50-**
**year window, 1-year step, right aligned) between DBI PC1 and the JJA mean temperature. The shaded area identifies the confidence**
**interval at 95 % (b). DBI PC1 and JJA mean temperature anomalies (grey line) and their Pearson correlation coefficient ($r$),**
**explained variance ($R^2$), and Spearman correlation coefficient ($\rho_s$) (c). The solid, dashed and dotted black lines in (a) and (b)**
**represent significance at the 0.05, 0.01 and 0.001 levels, respectively. (For interpretation of the references to colour in this figure**
**legend, the reader is referred to the web version of this article.)**

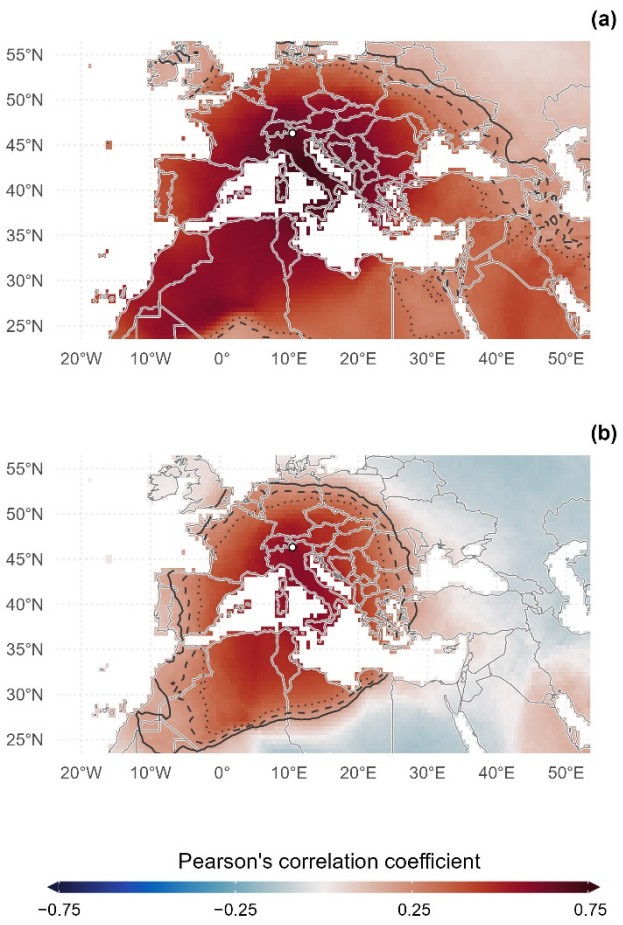

**Figure 4: Pearson's spatial correlation coefficient between DBI PC1 and CRU TS4.07 mean temperature for the period 1901–2013**
**for the raw series (a) and high-frequency domain (b). The solid, dashed and dotted black lines represent significance levels of 0.05,**
**0.01 and 0.001, respectively. The white dots represent the locations of the sampling stands. (For interpretation of the references to**
**colour in this figure legend, the reader is referred to the web version of this article.)**



### 4.2 Mean June–August temperature reconstruction

Starting from PC1 (ANBO+BARC+PALP), DBI-based JJA temperature reconstruction was attempted. The transfer function

was tested using the CCV and BTFS approaches, which returned stable results in the high-frequency domain, whereas the instability of the transfer function was highlighted for the row data and low-frequency domain (Table 2). The explained variances are similar in both calibration periods used, as are the RE values, which returned positive values with a limited standard deviation. In contrast, CE values were positive only for the high-frequency domain. The CCV results were corroborated by the results obtained by the more rigorous BTSF tests. Excluding the low-frequency domain, only in one case

(i.e., raw data intercept) did the calculated confidence interval at 95 % fail to pass the test, indicating a difference in offset between the two considered periods (i.e., calibration and validation). Similar results were also obtained for PC1 (ANBO+PALP) (Table S1 in the Supplementary Material). The ANBO chronology-based reconstruction, however, failed to pass the BTFS test for the slope ratio and explained variance ratio (Table S2 in the Supplementary Material). Finally, the Durbin–Watson test returned significant results for the low-frequency domain and the more recent half of the raw data,

highlighting a nonzero autocorrelation in the residuals of these models.

| | Cal. period [CE] | $R^2_{cal}$* | DW | CCV* | | BTFS† | | |
|---|---|---|---|---|---|---|---|---|
| | | | | RE | CE | Intercept-ratio | Slope-ratio | $R^2$-ratio |
| Raw | 1764–1888 | 0.42±0.08 | 1.71 | 0.42±0.01 | *–0.23±0.11* | *[0.888÷0.932]* | [0.584÷1.245] | [0.620÷1.922] |
| | 1889–2013 | 0.39±0.08 | *1.22* | 0.37±0.01 | *–0.28±0.16* | | | |
| Low | 1764–1888 | 0.64±0.06 | *0.04* | 0.63±0.01 | *–0.69±0.10* | *[0.905÷0.933]* | *[0.575÷0.866]* | [0.949÷1.732] |
| | 1889–2013 | 0.50±0.06 | *0.01* | 0.49±0.01 | *–1.92±0.50* | | | |
| High | 1764–1888 | 0.37±0.08 | 2.13 | 0.35±0.01 | 0.23±0.01 | [–8.707÷13.395] | [0.662÷1.845] | [0.672÷3.592] |
| | 1889–2013 | 0.25±0.07 | 2.18 | 0.23±0.02 | 0.34±0.02 | | | |

**Table 2: Explained variances of the calibration periods and statistical parameters of the CVV and BTFS procedures between PC1 (ANBO+BARC+PALP) and the JJA mean temperature. Italicized values identify parameters that do not pass the statistical tests (at the 95 % level when applicable). *One standard deviation is reported as a measure of uncertainty. †95 % confidence intervals are reported for BTSF parameters. For a detailed description of the BTFS parameters, please refer to Buras et al. 2017.**

Regarding the PC1 (ANBO+BARC+PALP) reconstruction (1730–2013 CE) based on raw data, periods of negative anomalies were shown in the 1740s–1760s, 1800s–1840s, 1880s, and 1960s–1970s (Fig. 5). However, the PC1 (ANBO+PALP) reconstruction (1611–2015 CE) reported negative anomalies in the 1620s–1630s and 1690s and, coherently with the previous 1750s, 1800s–1830s, 1880s, and 1960s–1970s (Fig. 5). The reconstruction performed considering only the ANBO chronology (1502–2015 CE) revealed negative anomalies in the 1500s–1520s, 1580s–1600s, 1620s–1670s, and as well as in other

proposed reconstructions in the 1690s–1710s, 1750s, 1790s–1850s, 1870s–1910s, and 1960s–1970s (Fig. 5). These cool periods were alternated by periods with positive anomalies that exceeded the symbolic threshold of +1 °C over the 1961–1990 mean, starting from the 2000s. More specifically, there were particular years in which the anomaly value exceeded two standard





deviations from the reconstruction anomaly mean (Table 3). Evidently, after the 1850s, the number of years with particularly negative anomalies rapidly decreased, whereas the number of years with exceptionally positive anomalies increased.

| Predictor | Negative anomaly year [CE] | Positive anomaly year [CE] |
|---|---|---|
| PC1 (ANBO+BARC+PALP) [1730–2013 CE] | 1753 1754 1813 1814 1815 1816 1819 1821 1830 1838 1880 1889 1956 1964 | 2003 2009 |
| PC1 (ANBO+PALP) [1611–2015 CE] | 1628 1633 1675 1753 1754 1813 1814 1815 1816 1819 1821 1830 1841 1880 1884 1909 1956 1964 | 2003 2005 2007 2009 2012 |
| ANBO [1502–2015 CE] | 1515 1524 1628 1633 1675 1699 1721 1753 1754 1813 1814 1815 1816 1819 1821 1830 1841 1888 1909 1956 | 1548 1994 2003 2005 2007 2009 2012 2013 |

**Table 3: particularly cold or warm years in which anomaly values differ by at least two standard deviations from the mean.**

## 5 Discussion

The data presented here represent, to our knowledge, the first attempt to use European larch BI data for dendroclimatic purposes in the European Alps. In fact, only one other study used BI data from this species but for dendroentomological purposes (Arbellay et al., 2018), whereas only a few studies with dendroclimatic purposes exist for the area but focused on other alpine species (Cerrato et al., 2023; Frank and Nicolussi, 2020; Österreicher et al., 2014; Trachsel et al., 2012; Babst et al., 2009). Consequently, this is also the first approach to dendroclimatic reconstruction based only on the BI of larches in the Southern Rhaetian Alps but also for the European Alps in general.

Regarding the different percentages of lighter (or darker) pixels, a total of 24 chronologies were obtained for their analysis. Preliminary analyses allowed us to observe slight differences in correlation values between the BI and climate data as a function of the considered pixel percentage (for both the EWBI and LWBI). The highest correlation was obtained for the DBI chronology derived from the difference between the mean of the pixels belonging to the first darker quartile (25 % of the LWBI pixels) and the mean of the pixels belonging to the whole earlywood (100 % of the EWBI pixels; not shown); thus, this chronology was used for successive analysis and considerations. Since the aim of this study was to verify the possibility of using BI data from European larch as a climate proxy, a detailed study on the mechanism that drove differences in correlation and thus the amount of explained variance as a predictor is worthy of future investigation but beyond the scope of this study. However, the highlighted slight influence of the considered percentage of pixels on the correlation with climate is corroborated by a previous study performed using Swiss stone pine (*Pinus cembra* L.), which attributed greater importance to the sampling polishing process than to the considered pixel amount or extractives removal process (Cerrato et al., 2023).

The considered DBI site chronologies showed highly coherent behaviour, confirmed by the high correlation values (Table 1), by both the high explained variance of the first principal component (82.32 %) and by the stability of the EPCA analysis (Fig. S2 in the Supplementary Material). Moreover, since the samples in this study were not washed in a Soxleth apparatus, it is





important to highlight that no evident offset between sapwood and heartwood sample data was noticeable in the DBI chronologies of all the sites (Fig. 2). This led to the speculation that, unlike for other species (e.g., Seftigen et al., 2020; Blake et al., 2020; Fuentes et al., 2018), the simple DBI chronology could be sufficient for obtaining a good and reliable climate

proxy for European larch, at least for samples collected from living individuals. This does not exclude the possibility that the removal of extractives, the adjustment of the BI values (Björklund et al., 2015), and/or the use of regional curve standardization (Helama et al., 2017) together (or not) with a signal-free approach (McPartland et al., 2020; Melvin and Briffa, 2008) will improve the already high regression performances.

### 5.1 BI–climate correlation

Static correlation analyses highlighted the strong influence of the summer (i.e., June–August) mean temperature on the PC1 DBI chronology (Fig. 3a). Similar but lower results were also obtained for the mean minimum and maximum temperatures. These high correlation values with June–August mean temperatures and even greater correlation with JJA mean temperatures indicate that the common signal observable in the chronologies is due to the influence of this specific climatic parameter on tree growth. These observations underline that BI responds to the same limiting factors that also affect other well-studied tree-

ring parameters (e.g., maximum wood density and total ring width) both in the area (Cerrato et al., 2018; Coppola et al., 2013) and in the Alps (Leonelli et al., 2016; Büntgen et al., 2011, 2006, 2005; Frank and Esper, 2005), even if spatial and/or physiological heterogeneity in the climate response within the species may exist (Saulnier et al., 2019; Carrer and Urbinati, 2004).

The moving correlation function between the PC1 DBI and mean temperature (Fig. 3b) highlights a relatively stable correlation

in the raw data and high-frequency domain, with an impactful decrease in the correlation coefficients in 1964 CE and 2003 CE. The decrease recorded since 1964 CE is imputable to a known outlier in the BI data (Fig. 3c). In fact, the 1964 CE was characterized by one of the worst larch budmoth (*Zeiraphera diniana* Gn.) outbreaks in the area (Baltensweiler and Rubli, 1999), as reported in a previous total ring-width analysis (Cerrato et al., 2019b; Turchin et al., 2003). Moreover, considering the occurrence of larch budmoth outbreaks in the area, most of the other particularly negative anomalies can be tentatively

associated with parasite outbreaks (e.g., 1956, 1909, 1888/89, 1884, 1880, 1841, 1838, 1830, 1821, 1753/54, 1721 CE; Table 3; Cerrato et al., 2019b; Arbellay et al., 2018; Büntgen et al., 2009). Conversely, the 2003 CE was influenced by the meteoclimatic conditions that occurred during that summer (Beniston, 2004; Fink et al., 2004). Despite the coherence shown by the DBI data and temperature time series in this year, denoting the relatively good performance of the DBI data for identifying even meteorological extremes, a sudden increase in the correlation coefficient (Fig. 3b) could be considered a

methodological artefact due to the effect of these outliers on the correlation values (Fig. S5 in the Supplementary Material). In fact, if the 1964 CE is removed, the sudden decrease is also removed, and the moving correlation values assume more linear behaviour (Fig. S5 in the Supplementary Material). Nevertheless, although the raw data correlation values do not show any monotonic trend, the correlation values between the DBI and JJA temperature, considering the whole correlation series in the





high-frequency domain, preserve their negative trend even if known outliers (i.e., 1964 and 2003 CE) are removed, but show

a significant upward trend if the trend analysis is focussed on 1960–2013 period (Mann-Kendall S: 379; p-value<0.01).

Previous analysis performed on TRW in the same study area (Cerrato et al., 2018; Coppola et al., 2013, 2012) reported the reduction in the correlation between tree-ring parameters and climate in the most recent portion of the chronologies. This is an issue known as the 'divergence problem' (D'Arrigo et al., 2008). The divergence problem is ubiquitous but aleatory in the Northern Hemisphere (Anchukaitis et al., 2017; Wilson et al., 2016; Büntgen et al., 2008), and it was observed especially when

total ring width was considered, whereas it was attenuated when other tree-ring parameters were used (Büntgen et al., 2006). Among the other, standardization method could be one sources of the divergence (D'Arrigo et al., 2008), however, applying best practice of detrending, this issue could be attenuated (Ols et al., 2023). In the study area, the results showed that the use of BI data attenuated the influence of divergence compared to the use of TRW chronologies (Cerrato et al., 2018; Coppola et al., 2012), as similarly reported for Swiss stone pines in the same area (Cerrato et al., 2023), for Norway spruce (Picea abies

(L.) H. Karst.) in the Carpathian Mountains (Buras et al., 2018) and for white spruce (*Picea glauca* (Moench) Voss) in Yukon, North America (Wilson et al., 2019). However, due to the continuity of the decrease highlighted by the high-frequency domain along the entire correlation series (even if a significant upward trend with values around $r = 0.50$ is appreciable in the last decades, especially if 1964 and 2003 CE outliers were removed; Fig. 3b and Fig. S5 in the Supplementary Material) and considering that, in most cases, the 'divergence problem' in the area started to affect the correlation with TRW in the 1960s

(Cerrato et al., 2018; Coppola et al., 2012), other factors apart standardization should be considered causes of the loss of explanatory power along the series, and probably not ascribable to the 'divergence problem' *sensu stricto*. Initially, we speculate that since a decrease in correlation affects the high-frequency domain and thus the quasiannual variation, the standardization method used should not have a relevant role. On the other hand, changes in climatic conditions, such as a more important role of precipitation quantity and distribution related to higher temperatures and thus an enhanced drought signal

embedded in the series, could be an alternative explanation for the observed correlation pattern. However, additional studies are needed to better understand the observed trends in correlation values. On the other hand, considering correlations coefficients, in the last decades, contrarily to the correlation values obtained between TRW and mean temperatures, the values increased, denoting the absence of the 'divergence problem' in the raw series and the influence of the low frequencies on correlation values.

Spatial correlations highlight that the BI data are representative not only of the local temperature but also, as expected, of the spatial homogeneity of this climatic parameter (Brunetti et al., 2006), which is also representative of a broader area (Fig. 4). Specifically, the raw data reveal high and significant values across an area spanning from the Mediterranean basin to the Middle East, North Africa and the Arabian Peninsula (Fig. 4a). However, this significant correlation and its wide spatial representativeness are not stable over time. In fact, starting from positive significant values calculated only over the

Mediterranean basin at the beginning of the time series, these values widely and rapidly expand towards the southeast, broadening the correlation significance in the last portion of the time series (Fig. S3, Animation S1 in the Supplementary Material). This enlargement is consistent with what is observed at the local site (Fig. 3b), where the raw data show a rapid



increase in correlation values, probably due to the retention of the low frequencies in the raw data. In fact, as highlighted by the moving correlation analysis, the low-frequency domain reaches its maximum correlation with the JJA temperature in the last portion of the analysed period, influencing the correlation of the raw data. This hypothesis is also corroborated by the results of the CCV, BTFS and Durbin-Watson tests, which revealed the nonstationarity of the regression intercept between the calibration and verification periods, which was even more pronounced in the low-frequency domain (Table 2). The high-frequency domain i) supports the hypothesis of the not-negligible influence of the low-frequency domain in the raw data, and ii) confirms the BI data as a potential proxy with an explanatory power higher than that reported by TRW for the same area (Cerrato et al., 2018; Coppola et al., 2013) and sometimes comparable to those obtained by MXD data (Büntgen et al., 2006), at least at high frequencies, as reported for other alpine species (Cerrato et al., 2023). Its spatial correlation is consistent with previous results, revealing a contraction of the spatial representativeness of the high-frequency band across the whole time series (Fig. S4, Animation S2 in the Supplementary Material), revealing the constant and significant negative trend shown by moving correlation analysis (Fig. 4b), even if stabilized in the last decades.

## 5.2 Mean June–August temperature reconstruction

The PC1s and the ANBO chronologies showed good predictive ability (Table 3; Tables S1 and S2 in the Supplementary Material), especially in high-frequency domain. All the tested chronologies had positive RE values, whereas CE values are positive in high-frequency domain and slightly negative considering the raw data. These results highlights the potential of DBI chronologies as proxies for the selected target climatic factor under certain condition. These results are corroborated by the more rigorous BTFS tests (Buras et al., 2017), where the PC1 raw chronologies exhibit instability in terms of the regression offset parameter (Table 3 and Table S1 in the Supplementary Material), implying weak nonstationarity in the trend (described by scenario I in Buras et al., 2017), whereas the ANBO-based reconstruction reveals nonstationarity in the regression slope and, as a consequence, in the explained variance, which is the cause of major trend instability (outlining scenarios II and III in Buras et al., 2017), and implies nonstationarity of the transfer function in the low-frequency domain. The nonstationarity of tree-ring proxies is widespread, and its presence in BI data has already been assessed in other species (e.g., Engelmann spruce, *Picea engelmannii* Parry ex Engelm. (Wilson et al., 2014), Scot pine, *Pinus sylvestris* L. (Rydval et al., 2016), and Swiss stone pine (Cerrato et al., 2023)), reflecting complex biological environmental-growth interactions (Wilmking et al., 2020) as well as complex interactions among abiotic entities (i.e., glaciers), biological growth and the environment (Cerrato et al., 2020; Leonelli et al., 2011). Analysis in the high-frequency domain, however, always passes the BTFS tests but at the expense of the retained low-frequency trend, which is relevant for climatic reconstructions (Esper et al., 2002), and must be reintegrated to obtain reliable long-term reconstructions (Rydval et al., 2016).

Based on these assumptions, raw data-based reconstructions have been performed considering PC1s and ANBO chronologies, supposing that a weak nonstationarity of the transfer function returned more useful information than those obtainable by the deletion of the entire low-frequency, at least at this preliminary stage of the study. A comparison between the reconstructed JJA temperature and both instrumental data (Fig. 5a) and previously proposed reconstructions (Fig. 5b–d) reveals good





agreement between the considered series. Generally, the DBI-based JJA temperature reconstruction proposed here returns completely comparable anomalies. The slight overestimation of the anomalies noticeable when the comparison with instrumental data or total ring width and maximum wood density-based reconstruction are considered (Fig. 5a and Fig. 5d), as exclusion of the more recent period, where an underestimation occurs. Nevertheless, local and multiproxy data (Fig. 5a–c)

corroborate the validity of the proposed reconstruction, agreeing well with respect to the trend and within the calculated RMSE. The highest concordance was observed with local data obtained by total ring width reconstruction (Coppola et al., 2013), which used chronologies from other valleys of one of the massifs sampled here (i.e., Adamello–Presanella), and from multiproxy-based reconstruction (Trachsel et al., 2012). Comparing the performance of the regression against the temperature of both the DBI and TRW data of ANBO, the only here used site chronology for which a TRW-based temperature reconstruction exists

(Cerrato et al., 2018), the regression involving DBI data returned higher indices and thus results better correlated (with higher predictive ability) with JJA temperatures than the latewood width with June–July mean temperatures (the meteorological variable that return the higher regression parameters considering the ring width). This reinforce the idea that DBI could be considered a better predictor than ring widths for summer temperatures also for European larch.

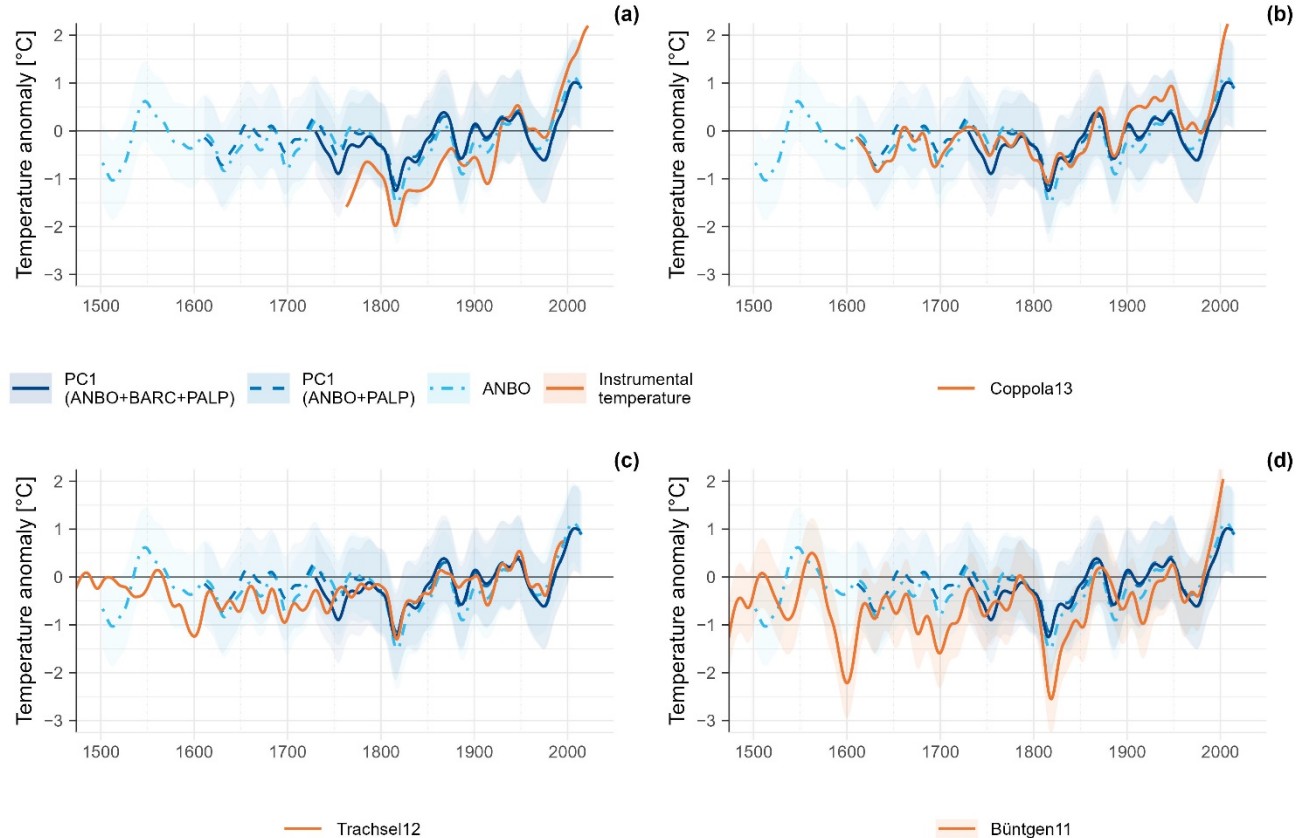





**Figure 5: Comparison between DBI PC1 and meteorological data (a) or the previously published JJA reconstruction (b–d). The solid, dashed and dotted blue lines are the JJA reconstructions performed considering the DBI PC1 of ANBO, BARC and PALP (1730–2013), ANBO and PALP (1611–2015), or only the ANBO chronology (1502–2015), respectively. The shaded area identifies the lower and upper confidence intervals at 95 %, where available. (For interpretation of the references to colour in this figure legend, the reader is referred to the web version of this article.)**

When analysing the PC1s-based reconstructed trends, these are corroborated by well-known climatic dynamics in the Alps. In fact, the cold phases reconstructed between the 17th and 19th centuries agree well with the known local acmes of the Little Ice Age (LIA), a period generally characterized by low temperature. During the LIA, the glaciers in the area, as well as along the entire alpine range, reach their Holocene maximum extension, even if asynchronously (Cerrato et al., 2020; Zemp et al., 2015; Carturan et al., 2014, 2013). Regarding the oldest portion of the ANBO raw-based reconstruction, even considering the major

trend non-stationarity highlighted by the BTFS tests (Table S2 in the Supplementary Material), the behaviour of the anomalies is very similar to the trend reported by total ring width-based reconstruction (Cerrato et al., 2018), and it is probably influenced by anthropogenic activities and wood harvesting and/or management performed in the area in the 16th century (Backmeroff, 2001).

After the coolest phases of the LIA, a progressive noncontinuous increase in the reconstructed JJA anomalies is evident and is

corroborated by other reconstructions and instrumental data (Fig. 5). In fact, JJA temperature anomalies started and continue to grow between the 1850s and today, with a major hiatus occurring during the 1970s–1980s. This latter cooling phase is corroborated by instrumental data, other reconstructions and environmental evidence that reported the last readvance of some glaciers in the area during this period (Salvatore et al., 2015). After the 1980s, the highest anomaly values of the entire series were reported, in accordance with more recent climate dynamic evidence (IPCC, 2018). Indeed, during this last phase, the

majority of the positive anomalies are identified (Table 3), and they almost correspond with years that are known for their exceptionally warm temperature across the European Alps (IPCC, 2018; Beniston, 2004; Fink et al., 2004).

## 6 Conclusions

In this paper, we focus on the utilization of European larch within the Southern Rhaetian Alps, demonstrating its potential for providing insights into the region's climate history. Specifically, we explored the application of blue intensity (BI), a relatively

novel technique, to obtain a proxy with predictive skills similar to those shown by using MXD data. In this context, the application of BI data analysis offers a promising tool for enhancing our understanding of past climate dynamics in the study area and regionally by providing additional information to those retrieved from other methodologies (e.g., TRW, MXD, wood anatomical traits and isotopes). In fact, the obtained results show that the BI data are representative of the mean JJA temperature at both the local and regional scales. The obtained data and their predictive ability are supported by the positive results obtained

by more rigorous tests of regression stationarity, highlighting the positive predictive ability of BI for European larch, as well as for other coniferous species already tested in the area and worldwide, at least in high-frequency domain. Some issues regarding the long-term trend persist and are worthy of future investigation. Moreover, opening the possibility of integrating

the use of BI data with more traditional dendrochronological techniques widely applied to European larch in the Alps, this study represents not only a first step towards promoting the use of BI data as a surrogate of MXD data in the European Alps

but also the possibility of obtaining MXD-like data from a greater number of laboratories for a highly investigated species to address critical questions related to the historical climate of the Southern Rhaetian Alps and the Alps in general.

## Author contribution

**Riccardo Cerrato**: Conceptualization (equal); Data curation (equal); Formal Analysis; Investigation (lead); Methodology

(lead); Project administration; Visualization; Writing – original draft. **Maria Cristina Salvatore**: Conceptualization (equal); Data curation (equal); Supervision (equal); Writing – review & editing (equal). **Michele Brunetti**: Data curation (equal); Methodology (supporting); Writing – review & editing (equal). **Andrea Somma**: Investigation (supporting). **Carlo Baroni**: Conceptualization (equal); Funding acquisition; Resources; Supervision (equal); Writing – review & editing (equal).

## Competing interests

Any use of trade, firm, or product names is for descriptive purposes only and does not imply endorsement by the in- volved universities. The authors declare that they have no conflict of or competing interests.

## Acknowledgements

This research was supported by the Laboratory of dendrochronology of the Department of Earth Sciences (University of Pisa). We are very grateful to Dr. Gino Delpero and Luca Colato, Custodi Forestali of Comune di Vermiglio, and to the staff of the

Stazione Forestale of the Comune di Ossana (TN) for helping in field activities and sampling. We wish to thank Dr. Fabio Angeli, responsible for the Ufficio Distrettuale Forestale di Malè (TN), for sampling permission.

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
