# Peer review of "Multicentury mean summer temperature variations in the Southern Rhaetian Alps reconstructed from *Larix decidua* blue intensity data"

_Climate of the Past, 2023_

## Author Response (AR1)

**Reviewer #1 comments**

The paper may be interesting, but it is not ready for publication. For example, Section 4, introducing the results, is almost impossible to read. There is inconsistency in terminology, the presentation of tables lacks care, there are awkward expressions in English, and many results are not shown. As such, I cannot evaluate the paper at this stage.

*Reply: We thank Reviewer #1 for having read our manuscript. We are working to improve the manuscript according to the supplied comments.*

In this section:

There is confusion between the terms "series" and "chronologies".

*Reply: We have further clarified these terms by using "core series", "individual series" and "site chronologies".*

I do not understand why there are four EWBI chronologies and four LWBI chronologies coming from three sites.

*Reply: The four EWBIs and the four LWBIs are derived from the different percentages of EW and LW pixels considered in the analysis for each site. Since there are no standard procedures for this species, various percentages of pixels within the frame (i.e., 25, 50, 75, and 100 %) were considered both for EW and LW, as stated on P5/L116-118 of the manuscript. However, we recognize that this concept was unclear and we recognize that a detailed study and specific scientific design are needed to explore this concept; thus, we have decided to remove this part. Further investigations will focus on this topic.*

Line 189: What does "remaining" mean?

*Reply: We have removed the unnecessary and confusing "remaining".*

The table is difficult to read because the second row in column 2 is for EPS*-related values, while in column 4, EPS*-related values are presented in square brackets. There is no consistency.

*Reply: The text has been revised as suggested to make it clearer and more consistent. The entire table has been restructured to improve readability.*

In the same table, there are additional lines labeled "PCA." I do not know why.

*Reply: Thank you for highlighting this inconsistency. We have edited the table for clarification: the inconsistent rows have been edited in accordance with what is reported in the text, and we have changed the column name from "Valley" to "Chronology code".*

The paragraph at line 194: To which figure/table do the results refer?

*Reply*: We have removed these methodological results since we believe that they deserve a dedicated paper.

Line 194: What does "from the difference" mean?

*Reply*: To obtain the DBI series and DBI chronologies, differences between the LWBI and EWBI series were determined. We have attempted to clarify this concept.

Line 200: Did you mean to say, "The DBI site chronologies showed similar decadal variabilities"? The term "trend" might be confusing.

*Reply*: Thank you for the suggestion. We have modified the text accordingly.

Line 201: What does "in quantity but not in time" mean?

*Reply*: The entire sentence has been removed; it is an unnecessary description of the chronology that can be easily inferred from Table 1 and Figure 2.

Line 202: Replace "duration" by "chronology length" which is more commonly used.

*Reply*: The entire sentence has been removed; it is an unnecessary description of the chronology that can be easily inferred from Table 1 and Figure 2.

"(not shown)" appearing twice in the section is excessive. Such results should be shown.

*Reply:* We have added these results to the supplementary material and integrated the Results section accordingly.

**Reviewer #2 comments**

Cerrato et al. explores the value of tree-ring blue intensity – a surrogate parameter for wood density - from European Larch in the Southern Rhaetian Alps as a proxy of past temperature. The use of blue intensity is still in an explorative state and therefore more tree species and sites are needed to bring this field forward. The idea of the current study is therefore compelling and should be of interest to the dendrochronological community and beyond. With this said, I find the state of the current manuscript to be far from being ready for publication. I am sorry to be negative, but parts of the manuscript are at present untidy, there are sometimes important details missing from the text and the meaning of some sentences and paragraphs is not comprehensible.

*Reply: We thank Reviewer #2 for having read our manuscript and for the encouraging words provided. We have been working to improve the manuscript according to the supplied comments.*

For example, the authors write that four EWBI and LWBI chronologies, respectively, are produced in the current work, but they do not give any explanation how this is done given that there are three sites that have been sampled.

*Reply: The four EWBIs and the four LWBIs are derived from the different percentages of EW and LW pixels considered in the analysis for each site. Since there are no standard procedures for this species, various percentages of pixels within the frame (i.e., 25, 50, 75, and 100 %) were considered both for EW and LW. However, we recognize that a detailed study and specific scientific design are needed to explore this concept; thus, we have decided to remove this part. Further investigations will focus on this topic.*

Also, how is it possible to get 16 deltaBI chronologies from four EWBI/LWBI datasets? The authors need to carefully check the manuscript for these inconsistencies or unclarities and correct or clarify before a proper review of the manuscript can be performed.

*Reply: The 16 DBI chronologies were derived from the permutation difference between the EWBI and LWBI series from which the chronologies were built. We have removed these methodological results since more detailed and appropriately designed experiments are needed.*

Also, the manuscript requires a careful language check.

*Reply: We have relied on the service offered by American Journal Experts (certificate validation no. E502-D45C-A2F2-D167-022P). We have reported your complaints to American Journal Experts, and we have asked for a new language check with different editors.*

When it comes to the methodological aspects of the study, I wonder why the authors have decided not to wash the samples in either ethanol or acetone as the standard praxis advocates. Resin and other impurities may pose serious implications for both high- and low-frequency variability in the resulting BI time- series, potentially obscuring the climate-related signal. This issue should be properly addressed in the manuscript and the methodology clearly motivated if the authors should choose to proceed with unwashed samples.

*Reply: As stated by the reviewer in the review opening, the BI is still in an explorative state, and thus, in the literature, sample processing is approached with different modalities. To our knowledge, there is no tested standard procedure for Larix decidua Mill., but we followed Wilson et al. 2014, 2019, 2021), who processed samples for BI analysis without washing them. Other authors followed the same approach, where samples were only polished (e.g., Dannenberg and Wise, 2016; Dolgova, 2016; Arbellay et al., 2018; Heeter et al.,*

*2020). On the other hand, other papers refer to samples treated with solvents (e.g., Campbell et al., 2007; Björklund et al., 2014, 2015, 2020; Babst et al., 2016; Brookhouse and Graham, 2016; Buckley et al., 2018; Buras et al., 2018; Akhmetzyanov et al., 2020; Cao et al., 2020; Blake et al., 2020; Davi et al., 2021). We discussed this point in the "Materials and methods" section and have tried to better underline it in the new version of the manuscript. In our opinion, one of the strengths and greatest originality of the paper we have presented lies in being the second to use the BI of Larix decidua and the first to use BI data from this species as a climate proxy. To our knowledge, there is no tested standard procedure to follow. However, our results indicate that a climate signal, at least in the high-frequency domain, is retained with high levels of significance, although the samples were not treated with solvents. We agree with the reviewer about the necessity of investigating the effect of the resin and removal of other impurities. In addition, we agree with the reviewer about the influence of the CooRecorder frame dimensions and percentage of retained pixels on the results. In our opinion, determining the effects that impurities in the cores can induce in measurement results deserves dedicated work with properly designed experiments. This is a very interesting point for further investigation, and we will focus on this point in a dedicated paper.*

I have listed some specific points below that require either clarification or rephrasing.

P1/ L13: Note that there are other proxies that have similar or higher temporal resolution than tree-rings.

**Reply**: *We have rephrased the sentence.*

P1/L18: "The results showed that BI data from European larch share greater variance with June–August mean temperatures than total ring width measurements." Would be informative to get some numbers here.

**Reply**: *We have quantified the improvement obtained using BI compared to TRW. The sentence is now as follows: "The results in the high-frequency domain showed that BI data from European larches explained up to 38.4 % (26.7–48.5 %) of the June–August mean temperature variance in the study area; this result is 70 % greater than the mean temperature variance percentages explained by total ring width measurements for the same period in the area."*

P1/L18: unclear what is meant by "temporal and spatial quite-stationarity" and "regression indices"??

**Reply**: *We have rephrased the sentence as follows: "Moreover, the correlation values between the BI data and June–August mean temperature are stable over time, ranging between 0.40 and 0.71 (mean value of 0.57), considering a moving window of 50 years, as well as spatial scale, with significant values over the western and central Mediterranean areas returned for all the considered time windows."*

P2/L45: "Among the climatic proxies that can be used (Trachsel et al., 2012), dendrochronology represents an excellent tool for reconstructing the climatic variations that occurred in the past." Dendrochronology is not a proxy but the science that is based on chronological dating of tree-rings. Please rephrase the sentence. Also, change "tree-ring-based dendroclimatology" in the next sentence. Dendrochronology is always based on tree-rings and it is thus redundant to state this.

**Reply**: *In accordance with Reviewer #2's comments, we have replaced "dendrochronology" with "dendrochronological data"; the next phrase has been rephrased to avoid pleonasms.*

P2/L59: "innovative methods such as maximum wood density (MXD)" Both TRW and MXD are conventional proxies that have been used in climate reconstructions for decades. I would therefore be careful saying that MXD is innovative or novel.

*Reply*: We agree with Reviewer #2 and have removed the word "innovative" to avoid confusion.

P3/L85: change to Mount Vióz, Mount Adamello and Mount Presanella

*Reply*: The highlighted terms have been changed according to the reviewer's comment.

P3/L88: I cannot see that the coordinates of the three sampling sites have been provided in the manuscript.

*Reply*: The coordinates of the sampling site have been added to the "Study area" section.

P4/L95/figure caption: "Numbers within square bracket indicate sample size" clarify if this refers to the number or cores or the number of trees?

*Reply*: The caption refers to the total number of sampled trees. The caption was clarified and revised according to other suggestions.

P4/L99: "The precipitation distribution reaches a minimum in winter and a maximum in summer, at 140.8 mm and 288.1 mm, … " clarify which month/months the precipitation totals are provided for, and also over which period.

*Reply*: The passage has been revised as follows: "The precipitation distribution reaches a minimum in winter (December–February) and a maximum in summer (June–August) at 172 mm and 292 mm, respectively; whereas the mean annual value is 1017 mm in the 1961–1990 period (Crespi et al., 2018; Carturan et al., 2012; Brunetti et al., 2006)." Differences in the values compared to those of the original version occur after the upgrade and reanalysis of the precipitation dataset.

P5/L108; "… to highlight the ring boundaries. They were then scanned at 3200 dpi …" Split sentence for better readability.

*Reply*: We have followed the suggestion of the reviewer.

P5/L117: Mention briefly why these Coorecorder settings were adopted when measuring BI? Were they the ones that gave the highest Rbar values, or were they simply arbitrarily selected?

*Reply*: The selected values were a compromise between both the sample and average tree-ring widths and the measurement necessities. A detailed study of the influence of the frame dimension on Rbar will be considered in the future. The paragraph has been revised as follows: "In this study, considering that cores with a diameter of 5.15 mm were involved, a frame width of 100 pixels (equal to 0.8 mm at 3200 dpi) was used to measure the minimum latewood BI (LWBI) and maximum earlywood BI (EWBI) values. Frame depths of 50 and 200 pixels (equal to 0.4 and 1.6 mm at 3200 dpi, respectively) were considered good compromises between the average wood structure width and the measurement necessities and were subsequently used for measuring the LWBI and EWBI, respectively."

P5/L122: Why 16 DBI datasats? What is the difference between these datasets? Please clarify. Also, it is a common praxis to wash the samples in alcohol or ethanol to remove the discoloration caused by resin and other impurities. A better explanation of why this step was omitted is required.

*Reply: As described in the "Materials and methods" section, the 16 DBI chronologies were derived from the differences between the four LWBI and the four EWBI series (while considering the 25, 50, 75 and 100 % of the pixels within the measuring frame applied to the cores, as described on P5/L116–118). From each LWBI series, all the EWBI series were subtracted, resulting in 16 DBI series for each core). However, we removed these methodological results since they deserve more specific investigation. Regarding resin and impurity removal, as stated by the reviewer in the review opening, the BI is still in an explorative state, and thus, in the literature, sample processing is approached with different modalities. To our knowledge, there is no tested standard procedure for Larix decidua Mill., but we followed Wilson et al. (2014, 2019, 2021), who processed samples for BI analysis without washing them. Other authors followed the same approach, where samples were only polished (e.g., Dannenberg and Wise, 2016; Dolgova, 2016; Arbellay et al., 2018; Heeter et al., 2020). On the other hand, other papers refer to samples treated with solvents (e.g., Campbell et al., 2007; Björklund et al., 2014, 2015, 2020; Babst et al., 2016; Brookhouse and Graham, 2016; Buckley et al., 2018; Buras et al., 2018; Akhmetzyanov et al., 2020; Cao et al., 2020; Blake et al., 2020; Davi et al., 2021). We discussed this point in the "Materials and methods" section, and we tried to clarify this point. In our opinion, one of the strengths and greatest originality of the paper we have presented lies in being the second to use the BI of Larix decidua and the first to use BI data from this species as a climate proxy.*

P5/L126: "BI sample series belonging to the same individual were averaged to create the individual BI series."

*Reply: We have changed "sample series" to "core series" for clarity. The sentence is now as follows: "After LBM correction, the BI core series belonging to the same tree were averaged to create the individual BI tree series."*

P5/L127: "Some individual BI series showed an age trend; thus, they were standardized using a modified negative exponential curve. If the modified negative exponential curve failed to fit the trend of the individual series, they were standardized using a negative or a horizontal line. "This sentence gives the impression that only some of the series were treated for age trends. If so, a better explanation is needed to why the standardization was not adopted universally, when wood density is known to have an age trend and is therefore commonly standardized before climate reconstruction.

*Reply: All the series were treated for age trends. The series that exhibited an exponential negative trend were standardised using an exponential negative curve. Following the procedure introduced in ARSTAN software (Cook 1985, Cook and Holmes 1986; Cook and Holmes 1999), if the negative exponential curve does not fit with the trend shown by the considered tree-ring series, the alternative standardisation method to attenuate the age trend is to use a negative slope or a horizontal line. In contrast, if the question refers to why some trees show a negative exponential trend in BI values that can be fit by a negative exponential equation whereas others do not, it likely depends on i) the environmental history of the tree (even if trees that were macroscopically damaged or lived in a dynamic geomorphological context were excluded from the sampling) or ii) the ability to reach (or not reach) the pith during the sampling (some individuals show hearthwood damaged by fungus or bacterial activity and thus only the most external portion was available/suitable for the analysis), but these are normal issues encountered and accounted for in the TRW series. However, we have attempted to clarify this concept. The sentence is as follows: "Then, individual BI tree series were standardized using a modified negative exponential curve or a linear regression (Cook and Holmes, 1999)."*

*Cook, E. R., 1985. A Time Series Approach to Tree-Ring Standardization. Ph.D. dissertation, University of Arizona, Tucson.*
*Cook, E. R., and Holmes, R. L., 1986: User's manual for program ARSTAN. In Holmes, R. L., and Adams, R. K. (eds.), Tree-Ring Chronologies of Western North America: California, Eastern Oregon, and Northern Great Basin. Tucson: Laboratory of Tree- Ring Research, University of Arizona, 50–56.*
*Cook E.R., Holmes R.L., 1999. Users Manual for Program ARSTAN., Tucson, Arizona.*

P5/L131: "The mid-low-frequency domain was obtained using the same filter as a low-pass filter." Unclear which filter the authors refer to.

*Reply: The sentence has been rephrased as follows: "Finally, to highlight the mid-low-frequency domain (sensu Melvin 2004), a low-pass Gaussian filter with a window length of 30 years and a sigma of 5 years was applied to the BI site chronologies. The high-frequency domain of the site chronologies was obtained as residuals from the low-pass filter."*

P5/L134: "… were considered for the site chronologies." (suggestion)

*Reply: Thank you for the suggestion; we have followed it.*

P6/L136: "representativeness of each chronology compared to an infinite hypothetical population" (suggestion)

*Reply: Thank you for the suggestion; we have followed it.*

P6 "3.2 the paragraph grouping in this section needs some correction (e.g., see line L158/159).

*Reply: We thank the reviewer for highlighting these typos. The divided paragraphs have been merged into one coherent paragraph describing the gridded dataset.*

L151: "… and interpolating the longest and homogenized meteorological series available for the Alpine region " this section would really benefit from more info. Which are the met series the authors are referring to? Where are they located, how far from the study sites and at which elevation? How representative are these data for the studied sites? What about the accuracy in the early part of the record – especially in regard to precipitation? In the introduction the authors mention that the reliability decreases prior to 1875. It is then appropriate to use these records as calibration targets?

*Reply: All the information concerning the met stations involved in climate information reconstruction is summarized in a figure reporting the spatial distribution of T and P stations around the site, the temporal evolution of data availability for T and P stations located within a radius of 150 km centred into the centroid of the three sites and their elevation distribution (Fig. 1). The same figure was added to the Supplementary Material.*
*The performances of the interpolation methodologies are described in the cited references (i.e., Brunetti 2012, Brunetti 2014, Crespi 2018 and Crespi et al 2021), as stated in the manuscript.*

[Figure]

***Figure 1***: *a) Spatial distribution of temperature stations. b) Spatial distribution of precipitation stations. Red dots indicate stations within 150 km of the centroid of the sampling sites (black circle). c) Temporal evolution of available stations within 150 km of the centroid of the sampling sites (red dots in panels a and b). d) Station distribution versus elevation.*

P6/L166: "high-resolution spatially continuous interpolated gridded dataset" information about the spatial and temporal resolution should be provided, as well as its temporal coverage. Also, which parameters were used from the CRU dataset?

***Reply***: *Details about the CRU-TS datasets used have been added. The added sentence is "In addition, to assess the spatial coherence of the dendroclimatic signal, high-resolution (0.5×0.5 degree lat-lon) monthly spatially continuous interpolated gridded datasets of minimum, maximum, mean temperature, and precipitation were used (Harris et al., 2020; Climate Research Unit Time-Series (CRU-TS), v. 4.07; last accessed 10 October 2023)."*

P7/L87: Please provide more details around the 24 BI chronologies that were obtained. It is not clear how and why 16 deltaBI chronologies were constructed. Also, as I understand three sites were sampled (fig. 1), but four EWBI and LWBI chronologies obtained? How is this possible?

***Reply:*** *The four EWBIs and the four LWBIs are derived from the different percentages of EW and LW pixels considered in the analysis for each site. Since there are no standard procedures for BI for this species, various percentages of pixels within the frame (i.e., 25, 50, 75, and 100 %) were considered for both EW and LW, as stated on P5/L116–118. The 16 DBI chronologies were derived from the differences between the four LWBI and the four EWBI series; from each LWBI series, all EWBI series were subtracted, resulting in 16 DBI series for each core from which the chronologies were built, as described in the "Materials and methods" section. However, we recognize that in the present form, this concept could not be clear since Reviewer #1 also highlights the same misunderstanding and more specific experiments are needed to clarify this point. Thus, we have removed these methodological results and plan to include them in future work.*

P//L187: "… the remaining chronologies …" not clear which remaining chronologies the authors refer to. Please clarify or rephrase the sentence.

*Reply: We have removed the unnecessary and confusing "remaining".*

P7/L189: "…whereas the mean interseries correlation increased by 0.05 at most" unclear, needs clarification.

*Reply: Thank you for the suggestion. The sentence has been rephrased as follows in the amended version of the manuscript: "From the other 59 individual BI tree series, considering the EPS, the BI values of 18931 rings spanning 514 years (i.e., 1502–2015 CE; Table 1) were obtained".*

P7/table 1:
unclear what "valley" in the table head is referring to.

*Reply: Thank you for highlighting the unclear column name. We edited the table to better explain it and changed "Valley" to "Chronology code".*

[EPS* $r\bar{r}$] and [EPS* $\rho\bar{\rho}$] may be misinterpreted as the EPS statistics multiplied by the interseries correlation coefficients (also, in the table caption it is referred to as *EPS and not EPS*).

*Reply: Thank you for the suggestion. The text has been revised as suggested (i.e., *EPS) to make it clearer and more consistent.*

Table captions are placed above and not below the table.

*Reply: Thank you for noting this. The table captions have been moved as requested.*

"correlation with other chronologies" which parameter and which period? Also, are the correlations performed on high-pass filtered data or using raw chronologies?

*Reply: Thank you for the suggestion. We have improved the caption as you requested.*

Why provide information on the number of trees that have not been used in this study?

*Reply: We agree that this information was unnecessary and have removed these numbers from the manuscript.*

P8/L202: "Moreover, the maximum sample depth of each chronology was comparable in quantity but not in time, with the duration reduced by approximately 100 years between ANBO and PALP and another 100 years between PALP and BARC" ?? Meaning unclear.

*Reply: Thank you for the suggestion. The sentence has been removed; it is an unnecessary description of the chronologies that can be easily inferred from Table 1 and Figure 2.*

P9/L214: "PC1 (ANBO+BARC+PALP)" unclear if we are still talking about deltaBI here?

*Reply: No, we are not referring to the DBI sensu strictu; we are referring to the first principal component resulting from the PCA applied to the three selected DBI chronologies (i.e., 25 % of the darkest pixels of the LWBI and 100 % of the pixels of the EWBI) of the study area. This section addresses P8/L203–204 in the previous version of the manuscript: "Due to the coherence shown by DBI chronologies, a PCA was performed considering all the chronologies to better preserve the common variability, likely related to climate (Seftigen et al., 2020)." This section also addresses P8/L206–209: "The PCA confirmed that the considered chronologies shared a large portion of the original variance. In fact, the first principal component (PC1) explained more than 80 % of the variance alone, and all the chronologies were strongly positively correlated with it. Thus, PC1 (ANBO+BARC+PALP) was selected to represent the areal chronology (see Fig. S2 in the Supplementary Material for further details). To better explain the concept, we have revised the paragraph as follows: "The three DBI chronologies showed similar decadal variabilities, with only BARC reporting a slightly positive trend over time (Fig. 2). Due to the coherence shown by the DBI chronologies, a PCA was performed to highlight the common patterns of variability and to evaluate their relationships with climate (Seftigen et al., 2020). The results showed that the first principal component (PC1) explained more than 80 % of the variance alone, and all the chronologies were strongly positively correlated with PC1 (see Fig. S4 in the Supplementary Material for further details). However, the correlation values obtained between the mean temperature and both the EWBI and LWBI PC1 (ANBO+BARC+PALP) corroborate the hypothesis that the discolouration due to the heartwood-sapwood transition affects the analysis (Fig. S5 in the Supplementary Material), whereas the DBI PC1 (ANBO+BARC+PALP) seems to not be affected by this issue (compare Fig. 3 and Fig. S5 in the Supplementary Material). Moreover, comparing the correlation coefficient obtained between the mean temperature and both DBI PC1 (ANBO+BARC+PALP) and site DBI chronologies shows that the former returned slightly greater values (Fig. S6 in the Supplementary Material). Thus, the DBI PC1 (ANBO+BARC+PALP), identified here as PC1 (ANBO+BARC+PALP), was selected to represent the mean areal chronology and was used for further analysis."*

P9/L220: it should be explained how the correlation confidence interval was obtained.

*Reply: Thank you for your suggestion. Details about the method used to calculate the confidence intervals have been added. The sentence appears as follows: "Correlation indices were calculated in the R-project statistical environment via the treeclim (Zang and Biondi, 2015) package. The bootstrapping procedure described in DENDROCLIM2002 (Biondi and Waikul, 2004) was applied to calculate the correlation indices and their 95 % confidence intervals via the percentile confidence interval method (Zang and Biondi, 2015; Dixon, 2001).".*

*Dixon P.M., 2001. Bootstrap Resampling. In: Encyclopedia of Environmetrics., Wiley. https://doi.org/10.1002/9780470057339.vab028*

P9/Sect. 4.1 What about the other BI parameters (EWBI and LWBI) and their climate signal imprint? Also, how is the BI signal compared to the TRW signal? If this is an exploratory study of BI from Larix (as stated in the introduction) then why do the authors limit the climate response analysis to just deltaBI?

*Reply: As stated in the Results section of the original version of the manuscript, "The analysis of the BI chronologies and their correlation with meteorological datasets revealed that the influence of the considered percentage of the LWBI darkest pixels, as well as that of the EWBI lightest pixels, slightly affected the results. However, in general, DBI values obtained from a small percentage of the darkest LWBI pixels (i.e., 25 %) and from a large percentage of the EWBI pixels (i.e., 100 %) yielded better results, considering the values of their correlation with meteorological data". Moreover, as stated in the "Materials and methods" section, "to correct the heartwood/sapwood discolouration that characterizes this species, Delta BI (DBI) datasets were calculated as differences between the LWBI and EWBI datasets and analysed (Björklund et al., 2015, 2014)." These are the main reasons that the EWBI and LWBI were not considered separately even if they were tested during the study. Nevertheless, we followed the reviewer's suggestion, and we added information in the manuscript and Supplementary Material reporting results for EWBI, LWBI and their PC1, demonstrating that the use of DBI for the European larch necessary to attenuate the discolouration issue that affects the samples at the heartwood-sapwood transition. Moreover, we added results from the TRW for comparison with the results from previous studies on the area (Coppola et al., 2012, 2013; Cerrato et al., 2018).*

P9/L241: what is meant by "raw spatial correlation"?

*Reply: We thank Reviewer #2 for highlighting this typo. The phrase "raw" was deleted. The sentence is now as follows: "In fact, after an initial decrease that limited the significant correlation values to the areas around the Mediterranean basin until the beginning of the 1990s, a strong and rapid increase was observed (Fig. S8 in the Supplementary Material)."*

P12/249 "what is meant by "static correlation"?

*Reply: This term is defined in Zang and Biondi (2015) to indicate the correlation performed on the entire considered period in contrast to 'moving', which is used on a moving window. We agree with Reviewer #2 that this term is not commonly used; thus, we removed it.*

*Zang C., Biondi F., 2015. treeclim: an R package for the numerical calibration of proxy-climate relationships. Ecography, 38 (4), 431–436. https://doi.org/10.1111/ecog.01335*

P12/figure 4 caption: it should be indicated over which season the temperature has been aggregated.

*Reply: Thank you for your suggestion. This information has been added to the caption in the amended version of the manuscript.*

P12/L266: "row data and low-frequency domain"?

*Reply: We corrected this typo.*

P15/L314: the authors argue that the deltaBI transformation of the data may be sufficient to mitigate the heartwood-sapwood discoloration issue. Is this just speculations, or has a comprehensive comparison between the deltaBI and LWBI actually been made to draw these conclusions?

*Reply: This is not a speculation, as this claim was tested in the first stage of the analysis. The significant attenuation of the difference between heartwood and sapwood in the DBI group compared to that in the EWBI and LWBI groups can be seen in Fig. 2 in the present file. We also added these results to the Supplementary Material as Fig. S3.*

I recommend the authors to work on streamlining the discussion and chiseling out the main message. The section is in its present form very long and often repeats what has already been stated in the results (for example L370 – spatial correlation analysis).

*Reply: We thank the referee for the comment. We have attempted to shorten the manuscript according to the reviewer's suggestion and focus the discussion on the most important message.*

[Figure]

*Figure 2: Mean raw BI series of latewood (LW), earlywood (EW) and DBI (delta) (thick solid black line). Solid grey thin lines identify the raw individual series. The red area identifies the portion of the series where the transition from heartwood to sapwood occurs. Notably, there was significant attenuation of the influence of sapwood on the DBI values compared to both EW and LW.*

---

## Author Response (AR2)

**Editor's comments**

After the secound round of reviews, I'm pleased to tell you that your paper entitled "Multicentury mean summer temperature variations in the Southern Rhaetian Alps reconstructed from Larix decidua blue intensity data" has just been accepted for publication in C.Past.

Both reviewers are positive and think your paper interesting as it is based on the Blue Intensity, a new method able to document summer temperature changes.

Only minor revisions are now needed (see both rapports below).

Could you please answer carefully to each point and send me the improved version of your paper? I recommend to to take into account the reviewers suggestions to improve the discussion part (better discuss the limitations of your approach, discussion on extraction) ; I strongly agree with the reviewer 1 about the fact that the paper is referring too many times to materials that are not inside the paper.

*Reply: Thank you for your message. We are pleased that our work has been accepted for publication in Climate of the Past. Below, we provide our responses to all the comments made by the second-round reviewer.*

*We have expanded the discussion on resin and extractive extraction, as well as the heartwood/sapwood transition. Additionally, we have reduced references to the Supplementary Material, although we believe that including as many results as possible provides a valuable data reference for future research. However, we now clarify in the text that establishing a standard protocol for BI on European larch is beyond the scope of this manuscript.*

*For these reasons, while the reported results are important for the manuscript and future readers, they are not directly focused on the main topic of the text but serve as fundamental steps to support the protocol used. Furthermore, including these results in the main text would detract from the primary narrative and reduce the overall focus, especially considering that some of these additions were specifically requested by an anonymous reviewer during the first round of revision.*

*Best regards,*
*dr. Riccardo Cerrato on behalf of all authors*

**Anonymous Reviewer #3 – report #1 comments**

I critically read the paper of Cerrato et al., and I found the paper interesting, opening new possibilities in dendroclimatic reconstructions in the EU Alps based on European larch. It's true that many analyses tend to produce many graphs and possibilities of interesting deepening into data, however I personally do not like the supplementary material philosophy, and in my opinion the paper is referring too many times to materials that are not inside the paper. I understand this is a trending way, also supported by the journals.

*Reply: We thank the Reviewer for the kind comment. Regarding the amount of data included in the Supplementary Materials, as the Reviewer noted, "many analyses tend to produce many graphs and possibilities of interesting deepening into data." This is particularly true for studies that aim to apply a relatively new method to a new species and observe its response. However, we believe that including all the analyses in the main manuscript would make it excessively long and less focused. Moreover, as Dr. Björklund noted in one of his comments, we have made an effort to be as transparent as possible, which may occasionally come across as tedious. Nonetheless, we consider this transparency essential, especially because more studies on the BI of European larch are needed. Reporting as many results as possible provides a valuable data reference for future research, even though establishing a standard protocol for BI on European larch is beyond the scope of this manuscript.*

*That said, we have made an effort to limit and reduce references to the supplementary materials throughout the text wherever possible.*

Besides this consideration, I have some minor indications to provide to the Authors:

1) A network of 3 site chronology at maximum distance of approximately 8 km, lead to a reconstruction of climate variability over thousands of km around. Sites were selected along a modest altitudinal gradient, line94. Before passing to PC1, did you check if you had different climate-BI responses separately at the three sites? Please better discuss.

*Reply: Yes, we checked, and the results are reported in Figure S6 of the Supplementary Material. As shown, all three sites exhibit a positive response to June, July, and August mean temperatures, with the strongest correlations observed with June–August (JJA) mean temperature. In two out of three cases, the correlation between PC1 and JJA mean temperature is higher than that observed for the individual site chronologies (i.e., Pearson's r increases for BARC by 0.09, and for PALP by 0.16). However, when comparing the correlations of ANBO with JJA mean temperature to those of PC1, the latter is slightly lower by 0.09.*

*We have updated the discussion to reference this analysis. The revised first sentence of paragraph 5.1 now reads: "The correlation analyses highlighted the strong influence of the summer (i.e., June to August) monthly mean temperature on the individual site chronologies and especially on the PC1 (ANBO+BARC+PALP) chronology (Fig. 3a)."*

*Regarding the observation that chronologies from three sites located within a few kilometres in the Alps show a consistent positive response to regional-scale temperature variations, this is an expected result. This consistency is attributed to: i) the spatially homogeneous behavior of temperature; and ii) similar findings for other species in different parts of the world (e.g., Zheng et al., 2023; Seftigen et al., 2020; Heeter et al., 2019; Wilson et al., 2017).*

Heeter, K. J., Harley, G. L., Van De Gevel, S. L., and White, P. B.: Blue intensity as a temperature proxy in the eastern United States: A pilot study from a southern disjunct population of Picea rubens (Sarg.), Dendrochronologia, 55, 105–109, https://doi.org/10.1016/j.dendro.2019.04.010, 2019.

Seftigen, K., Fuentes, M., Ljungqvist, F. C., and Björklund, J.: Using Blue Intensity from drought-sensitive Pinus sylvestris in Fennoscandia to improve reconstruction of past hydroclimate variability, Clim. Dyn., 55, 579–594, https://doi.org/10.1007/s00382-020-05287-2, 2020.

Wilson, R., Wilson, D., Rydval, M., Crone, A., Büntgen, U., Clark, S., Ehmer, J., Forbes, E., Fuentes, M., Gunnarson, B. E., Linderholm, H. W., Nicolussi, K., Wood, C. V., and Mills, C.: Facilitating tree-ring dating of historic conifer timbers using Blue Intensity, J. Archaeol. Sci., 78, 99–111, https://doi.org/10.1016/j.jas.2016.11.011, 2017.

Zheng, Y., Shen, H., Abernethy, R., and Wilson, R.: Experiments of the efficacy of tree ring blue intensity as a climate proxy in central and western China, Biogeosciences, 20, 3481–3490, https://doi.org/10.5194/bg-20-3481-2023, 2023.

2) Working on PC1 then of course cut-off parts of the different climatic influences at the site level. Fig. 3: I see colored bars (raw, low and high –'frequency domain' could be written somewhere), I see here and there the gray bars, but what are the white bars depicting?

**Reply:** *We agree with Reviewer #3 that the use of PCA reduces the ability to distinguish the specific climatic influences at each site level. However, as noted by the Reviewer #3 in the previous comment, the sites are geographically close to one another. Therefore, since they are all sensitive to the same climatic period (i.e., JJA), the primary differences we can hypothesize between the sites are: i) a potential altitudinal gradient that influences the response of the trees at the different stands. It should be noted, however, that this gradient is limited, as also mentioned by Reviewer #3 in the previous comment; and ii) the effect of outbreaks of Zeiraphera diniana Gn., which affect European larch in the European Alps in a non-synchronous manner (Bjørnstad et al., 2002). Furthermore, the impact of defoliation due to these outbreaks is altitude-dependent, affecting the three sites differently (Cerrato et al., 2019). Thus, the application of PCA likely retains, in the first principal component, the factor that most strongly influences the dataset's variance. Based on our results, this is most likely JJA temperature.*

*Regarding the white bars, they represent the correlations that does not reach the 0.05 level of significance.*

Bjørnstad, O. N., Peltonen, M., Liebhold, A. M., and Baltensweiler, W.: Waves of larch budmoth outbreaks in the European alps, Science, 298, 1020–1023, https://doi.org/10.1126/science.1075182, 2002.

Cerrato, R., Cherubini, P., Büntgen, U., Coppola, A., Salvatore, M. C., and Baroni, C.: Tree-ring-based reconstruction of larch budmoth outbreaks in the Central Italian Alps since 1774 CE, iForest - Biogeosciences For., 12, 289–296, https://doi.org/10.3832/ifor2533-012, 2019.

3) Please better discuss the discoloration issue. As far I can see, the strong declining trend in the sapwood corresponds to a positive trend in the recent period for DBI that perfectly fits with the recent period of temperature rise. It is true that other authors found DBI effective, but since this is a climatic reconstruction, I would have personally excluded this latter period from the model (and also from the sensitivity analysis) to avoid the possible inclusion of biases. If the same trees will be sampled in 20-30 years, the areas now in sapwood will be included in the heartwood: would the nice positive trend disappear? I am not asking to go back to see how the reconstruction is affected without the most recent period, however the Authors should better discuss what limitation could face the proposed reconstruction.

*Reply: We agree with the Reviewer #3's comment. Heartwood/sapwood transition could be an issue but currently it is impossible to know what will happen in the future. However, we account for this in the revised version of the manuscript by adding a paragraph in the section 5.2. Now the manuscript reads as follows: "Regarding the recent observed trend, it is important to note that it coincides with a period characterized by the heartwood/sapwood transition in most of the samples used. Although the use of the DBI mitigates the discoloration issue, the notably good agreement between the BI and meteorological data may still be influenced by these conditions. However, it is currently not possible to determine whether, in the future, when the current sapwood will become part of the heartwood, the strong correspondence between the proposed proxy and the meteorological data will persist.".*

4) Especially because the Authors propose reconstructions based on different combinations of site chronologies (up to including only one chronology, ANBO; Fig. 5), I strongly recommend to include the sensitivity analysis for each site chronology separately.

*Reply: The analysis has been performed and reported in the Supplementary material as Figure S6. The motivation that drove this choice is based on the evidence that the PC1 resulted to be more sensitive to climate in two out of three cases (BARC and PALP). This is probably due to the fact that BARC and PALP are located at lower altitudes and thus probably more prone to larch budmoth outbreaks. The higher number of outbreaks and the higher severity could partially hamper the climatic signal inside these chronologies (even if procedures to try to attenuate this effect has been applied, as described in Materials and Methods section). However, albeit the LBM fingerprint, the chronologies of these valley still retain a significant climatic signal (alpha = 0.99). Thus, as explained in a previous reply, we decide to focus the narration in the manuscript on the more important results, but to be transparent with other researcher, we decide to include as many results as possible in the Supplementary Material providing a valuable data reference for future research*

**Dr. Björklund Jesper's comments**

This study expands the research using Blue Intensity to a new species, Larix decidua, growing in the Rhaetian Alps. New evidence from BI is always interesting, and new data on past climate are useful to consolidate current understanding of the climate system and climate change or if it challenges conventional understanding. The study is pleasant to read and uses for the most part standard protocol methods. Considering that it is a study using BI to explore climate, there may be a shortage of protocol experimentation, because BI is not state-of-the-art but an affordable shortcut to comparable information. I specifically think of the lack of resin extraction of the samples, which may or may not be influential on the conclusions. That said, it is a nice study that only needs some minor revisions in my view. Below there are some comments that might be useful in this process.

*Reply: We thanks Dr. Björklund for his kind comments. We acknowledge the limited protocol experimentation in the present study. However, to our knowledge, this is one of the first studies to use European larch BI for climatic reconstruction purposes. We believe it was important to demonstrate that BI performs well with this species in the European Alps, as this could encourage further research on European larch. We agree that more detailed methodological studies are necessary to fine-tune the protocol, obtain cleaner data, and achieve more accurate reconstructions from European larches.*

L67-70 This sentence is too dense, split up in two or three sentences. The terms "spectral analysis" and "nonlinearity" are used in ways that do not make them easy to understand for a reader not intimately familiar with the cited literature.

*Reply: We followed the suggestion of Dr. Björklund and modify the text as follows: "In fact, BI data, derived from the reflected-light spectral analysis of tree-ring samples, provide climatic information that is virtually identical to that acquired through MXD. The BI and MXD data reflect cell wall dimension rather than the TRW or cell wall compounds, and thus show strong similarities in terms of temperature correlation strength and autocorrelation (Björklund et al., 2021; Ljungqvist et al., 2020)."*

L120-130 The description of the method here is exemplary, but please consider the terminology for BI proposed in Björklund et al 2024. The problem highlighted in this paper is that over the past decade(s), LWBI and EWBI have been synonymous with both inversion states. For future research it would be simpler if BI was dedicated for data positively correlated with density, and that Blue Reflectance (BR), i.e., the raw state, would be dedicated to the non-inverted state, which is negatively correlated with density.

*Reply: We thank Dr. Björklund for highlighting this inconsistency. At the time of the initial submission, their work had not yet been published. We are pleased to revise the text in this paragraph to align with the proposed standard terminology. The revised text now reads as follows: "In this study, considering that cores with a diameter of 5.15 mm were involved, a frame width of 100 pixels (equal to 0.8 mm at 3200 dpi) was used to measure the minimum latewood Blue Reflectance (LWBR) and maximum earlywood Blue Reflectance (EWBR) values. Frame depths of 50 and 200 pixels (corresponding to 0.4 mm and 1.6 mm at 3200 dpi, respectively) were determined to be optimal compromises between the average wood structure width and measurement requirements. These frame depths were subsequently employed for measuring the*

*LWBR and EWBR, respectively. The offsets of the frame were set at 5 and –2 pixels for the LWBR and EWBR, respectively (Fig. S1 in the Supplementary Material). For the LWBR measurements, we considered the mean values of the 25 % of the darkest pixels in the frame, whereas all the pixels within the frame were considered for the EWBRI measurements (Cerrato et al., 2023). For easier comparison with climate data, BR values were inverted following standard procedures to derive BI values (Rydval et al., 2014; Wilson et al., 2014), consistent with the '2024 BI standard terminology' (Björklund et al., 2024)."*

L130 perhaps replace "devise different solutions" with "require attention"
***Reply****: We accepted the proposed changes.*

L135 Perhaps: Visual and statistical crossdating of ring width, from the core samples, ensured that all obtained BI-based values were also correctly crossdated
***Reply****: We accepted the suggestion modify the text according to it.*

L200-201 perhaps also check out Esper, J., Frank, D. C., Wilson, R. J., & Briffa, K. R. (2005). Effect of scaling and regression on reconstructed temperature amplitude for the past millennium. Geophysical Research Letters, 32(7). The regression deflates the variance, but scaling inflates the error.
***Reply****: We thank for the suggestion. We better explain the applied methodology that is a combination of regression and scaling. Text now reads as follows: "Then, the mean and the variance of the regressed DBI z-scores data were adjusted against the instrumental targets to avoid the typical loss of amplitude due to regression error (Carrer et al., 2023) and reducing the inflated error variance observed when only the scaling approach was applied (Esper et al., 2005).".*

L206-208 It is quite a large negative slope in the sapwood transition zone. Probably in part due to that ethanol extraction was omitted. It is very likely that EWBI and LWBI are affected by this transition, but is the effect completely neutralized in the delta parameter? This is difficult to say because the EWBI and LWBI have other calendar dated variances as well (climate, moths, stand dynamics..?). An initial experiment to test this could be to align all LWBI and EWBI on HW/SW transition date, and compare mean values before and after the transition. To facilitate the analysis further, the mean values of LWBI and EWBI can be set to zero in the HW zone by means of subtracting the HW mean from the timeseries. If the SW means of the LWBI and EWBI are the same, the effect is neutralized. However, since all HW/SW transitions occur at roughly the same years, this analysis will not completely neutralize the climate and other variances, so it will not be a perfect analysis. Note that I do not insist on such an analysis, but it could be informative as a test.
***Reply****: We followed the suggestion of Dr. Björklund and perform the proposed analysis on a single valley (ANBO). Preliminary results show that the creation of DBI only mitigate the effect of the HW/SW, thus specific designed scientific investigations are needed but are beyond the main aims of the present*

*manuscript. This is now highlighted in the discussion also in reply to a following comment trying to specify that even if in the present study the resin extraction was not performed it does not mean that it is not necessary.*

L230-231 seems like something is missing here, perhaps add an "and" correlates stronger with…? .."and does not correlate SIGNIFICANTLY with".. ?
***Reply***: *We modified the text as suggested.*

L251 time series?
***Reply***: *We replaced temporal series with the suggested time series.*

L252 DBI?
***Reply***: *We modify the text as suggested.*

Figure 3 A moving window correlation analysis with smoothed data may not be so statistically sound. At least it will be very difficult to reach significance after the loss of degrees of freedom is considered (even when the correlation is close to 1). I would remove this from panel b and state that such an analysis will not be so informative, it is enough with the visual comparison in panel c.
***Reply***: *We followed the Dr. Björklund's suggestion and modified the Figs. 3, S5, S6, and S7 accordingly. The text was modified between line 246 and 249 and now reads as follows: "The moving window analysis between PC1 (ANBO+BARC+PALP) and the JJA mean temperature revealed significant and stable correlation values at the 0.01 level when considering the raw data and high-frequency domain (Fig. 3b). Regarding the low-frequency domain, interpreting its significance is challenging, particularly given the loss of degrees of freedom caused by the applied smoothing function."*

Table 2 I appreciate the effort for transparency but I do not think it is necessary to include the regression tests for high-and low frequency filtered data.
***Reply***: *We partially agree with Dr. Björklund's comment. Since the high-frequency has been discussed in the section 5, we decided to remove from the Table 2 only the analysis regarding the low-frequency domain, according to what was done for Fig. 3.*

L322-340 The discussion on extraction is necessary because this study makes a step away from common practice. I think the authors have a time series which is sound, but I would be surprised if it would not be

slightly improved if resin extraction was included (especially in the high frequency rbar, and perhaps in the low frequency as well). Abundant resin can be very local and might not even be comparable in earlywood and latewood.

Perhaps something like this is visible in ANBO EWBI around 1800 CE in Figure S3. Probably this is something else, but in principle, this type of erratic behaviour could perhaps be mitigated with resin extraction? The discussion concludes that DBI is effective in mitigating HW/SW differences, which is fine. But surely the authors do not discourage from using resin extraction? Can a sentence or two be dedicated to this?

**Reply**: *We agree with Dr. Björklund and added the following paragraph at the end of the section: "Although the data collected for this study suggest that the creation and use of DBI significantly mitigate discoloration issues occurring at the heartwood/sapwood transitions in European larch samples, this does not imply that the removal of resins and extractives is unnecessary. Instead, it underscores the need for further analysis because no information is currently available on how resins and extractives affect the EWBI and LWBI timeseries derived from European larch samples. Additionally, resin abundance can vary greatly at a local scale and may not be comparable between earlywood and latewood, and therefore it influences the DBI time-series. To better understand the influence of resins and extractives on BI – both in heartwood, sapwood, and their transition zone – a specifically designed scientific investigation is required."*